# Exploring Porcine Precision-Cut Kidney Slices as a Model for Transplant-Related Ischemia-Reperfusion Injury

**L. Annick van Furth** [1,*] , **Henri G. D. Leuvenink** [1] , **Lorina Seras** [2] , **Inge A. M. de Graaf** [2] , **Peter Olinga** [3]
and **L. Leonie van Leeuwen** [1]

1    Department of Surgery, University Medical Center Groningen, University of Groningen,
9713 GZ Groningen, The Netherlands; h.g.d.leuvenink@umcg.nl (H.G.D.L.);
l.l.van.leeuwen@umcg.nl (L.L.v.L.)

2    Department of Pharmacokinetics, Toxicology and Targeting, Groningen Research Institute of Pharmacy,
University of Groningen, 9713 AV Groningen, The Netherlands; seraslorina@gmail.com (L.S.);
i.a.m.de.graaf@rug.nl (I.A.M.d.G.)

3    Department of Pharmaceutical Technology and Biopharmacy, Groningen Research Institute of Pharmacy,
University of Groningen, 9713 AV Groningen, The Netherlands; p.olinga@rug.nl

*    Correspondence: l.a.van.furth@umcg.nl; Tel.: +31-641638348

**Abstract:** Marginal donor kidneys are more likely to develop ischemia-reperfusion injury (IRI), resulting in inferior long-term outcomes. Perfusion techniques are used to attenuate IRI and improve graft quality. However, machine perfusion is still in its infancy, and more research is required for optimal conditions and potential repairing therapies. Experimental machine perfusion using porcine kidneys is a great way to investigate transplant-related IRI, but these experiments are costly and time-consuming. Therefore, an intermediate model to study IRI would be of great value. We developed a precision-cut kidney slice (PCKS) model that resembles ischemia-reperfusion and provides opportunities for studying multiple interventions simultaneously. Porcine kidneys were procured from a local slaughterhouse, exposed to 30 min of warm ischemia, and cold preserved. Subsequently, PCKS were prepared and incubated under various conditions. Adenosine triphosphate (ATP) levels and histological tissue integrity were assessed for renal viability and injury. Slicing did not influence tissue viability, and PCKS remained viable up to 72 h incubation with significantly increased ATP levels. Hypothermic and normothermic incubation led to significantly higher ATP levels than baseline. William's medium E supplemented with Ciprofloxacin (and Amphotericin-B) provided the most beneficial condition for incubation of porcine PCKS. The porcine PCKS model can be used for studying transplant IRI.

**Keywords:** kidney transplantation; donation after circulatory death; precision-cut kidney slices; normothermic machine perfusion

## 1. Introduction

Due to the worldwide shortage of donor organs, marginal kidneys from extended criteria donors (ECD) and donation-after-circulatory-death (DCD) donors are increasingly used for transplantation. These kidneys are exposed to more (ischemic) injury compared to kidneys from donation-after-brain-death (DBD) donors and are therefore more likely to develop ischemia-reperfusion injury (IRI) [1,2]. Ischemia is characterized by restricted blood supply, causing a shortage of oxygen delivery to the cells needed for adenosine triphosphate (ATP) production [3,4]. Upon reperfusion, the blood supply is restored, which induces an inflammatory response. The re-introduction of oxygen and normalization of pH are detrimental to the previously ischemic cells, leading to increased reactive oxygen species (ROS) production and thus oxidative stress [5]. Unfortunately, IRI's underlying pathophysiological mechanisms can lead to transplant-related complications, such as the development of delayed graft function (DGF), early graft failure, chronic allograft

nephropathy, and late graft failure [6,7]. Therefore, diminishing the effects of IRI would be of great interest.

Machine perfusion (MP) is a novel technique widely used to attenuate IRI. It provides the unique opportunity to preserve and resuscitate the donor organ outside the body [8,9]. MP can be performed at different temperatures and for various purposes. Hypothermic machine perfusion (HMP), with preservation at 4 °C, has been developed as an alternative to static cold storage (SCS), as it offers superior organ preservation, especially for grafts of suboptimal quality [10,11]. HMP can be performed with or without oxygen; however, recent studies show that adding oxygen during HMP results in better preservation of the mitochondria and fewer post-transplant complications [12–14]. Machine perfusion can also be performed at a physiological temperature through normothermic machine perfusion (NMP). NMP is performed with an oxygenated blood-based solution at 37 °C, mimicking physiological conditions to support metabolic activity and organ functionality [15]. To study machine-perfusion-related research questions, porcine kidneys are often used, as they are anatomically and functionality-wise similar to human kidneys [16]. Although NMP is widely studied to better preserve, repair, and assess the function and quality of the kidney graft [17–19], the perfusate composition and machine settings differ significantly between NMP protocols. The exact metabolic needs of an isolated kidney are still not fully characterized, and therefore, the appropriate parameters to assess renal function are not yet defined [20,21]. Additionally, NMP alone might not be enough to repair ischemic damage [22], and pharmaceutical interventions may enhance NMP significantly. Hence, more studies on transplant-related IRI and experimental NMP are essential.

NMP experiments are costly and time-consuming. Thus, an intermediate model for transplant-related research would be pivotal. The precision-cut kidney slice (PCKS) model could serve as such. PCKS are viable explants that can be used to study organ-specific cellular mechanisms, as they maintain cellular heterogeneity and organ structure [23]. PCKS can be prepared swiftly and in a reproducible manner, as the technical aspects are forthright. They have already proven their success in various research fields [24,25]. Ischemia-reperfusion studies have been performed with intestinal precision-cut tissue slices [26]; however, tissue slices have not been applied to renal transplant-related ischemia-reperfusion research.

As an almost infinitive number of PCKS can be prepared from just a single kidney, and as these slices can be incubated under normothermic and oxygenated conditions, we believe that this model would be suitable for studying IRI and ex vivo reperfusion. Therefore, the aim of this study was to investigate whether the PCKS model in combination with cold preservation is suitable as an ischemia-reperfusion model for transplant-related research. This study shows that oxygenated HMP leads to viable porcine PCKS for up to 48 h under both normothermic and hypothermic conditions.

## 2. Materials and Methods

### 2.1. Experimental Design

The experimental workflow is illustrated in Figure 1.

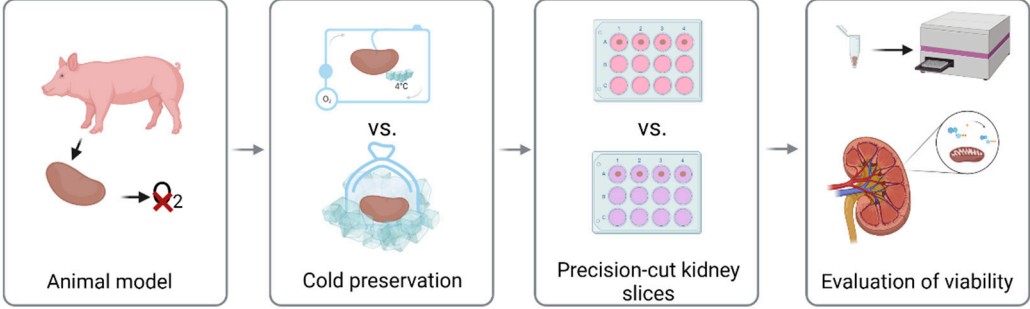

**Figure 1.** Experimental design of porcine precision-cut kidney slices.

## 2.2. Animal Model

All experiments were carried out with porcine kidneys. These kidneys were retrieved from a local slaughterhouse after a highly standardized slaughtering process. Pigs were anesthetized using an electric shock followed by exsanguination. The kidney was selected based on color, arterial branching, and the absence of abnormalities. Each kidney was exposed to 30 min of warm ischemia, thus exposed to the same extent of induced ischemic injury. This workflow was chosen to reflect the donation-after-circulatory-death conditions.

## 2.3. Cold Preservation

Kidneys were preserved and transported using SCS or HMP. All kidneys were flushed with 180 mL of 0.9% saline solution (Fresenius Kabi, Louviers, France) before cold preservation.

Kidneys preserved using SCS were submerged in 300 mL ice-cold University of Wisconsin (UW) cold storage (CS) solution (Bridge to Life Ltd., London, UK) and preserved on ice at 4 °C for 24 h before undergoing the slicing procedure.

Kidneys preserved using HMP were surgically prepared and connected to the Kidney Assist Portable (XVIVO, Gothenburg, Sweden) perfusion machine. HMP was performed for either 3 or 24 h at a set mean pressure of 25 mmHg using 330 mL oxygenated (100 mL/min) UW machine perfusion (MP) or CS solution at 3–5 °C.

## 2.4. Precision-Cut Kidney Slices

After cold preservation, kidneys were immediately flushed with 120 mL 0.9% saline solution to remove the UW solution from the vasculature. With a 10 mm blade, the renal capsule was carefully removed, and the cortex was cut off from the kidney. Cores were prepared from the cortex with a 6 mm biopsy punch and stored in ice-cold UW-CS solution.

PCKS were obtained using the Krumdieck tissue slicer (Alabama Research and Development, Munford, KY, USA). The Krumdieck slicer was assembled and filled with ice-cold oxygenated Krebs–Henseleit buffer (25 mM NaHCO$_3$ (B. Braun, Melsungen, Germany), 25 mM D-glucose (Merck, Darmstadt, Germany), and 10 mM HEPES (Merck, Darmstadt, Germany), pH: 7.4). Slices were collected and selected on round and intact macroscopic morphology. The slices weighed 4.5–5.5 mg, with an estimated thickness of 300 μm. To ensure viability, the slices were preserved in ice-cold UW-CS solution for a maximum of one hour. Slices were then incubated for 24, 48, and 72 h at 4 °C or at 37 °C with 80% O$_2$ and 5% CO$_2$ while gently shaking at a rate of 90 rpm. The medium composition for each experimental group is described in Table 1 and was refreshed every 24 h.

**Table 1.** Incubation conditions.

| Experimental Group | Medium Composition |
|---|---|
| WME | Williams Medium E (1X) with GlutaMAX (WME) (Gibco) + 10 μG/mL ciprofloxacin (Fresenius Kabi, France) |
| WME + Glucose | WME + 10 μG/mL ciprofloxacin + extra added D-(+)-glucose solution (Sigma-Aldrich, St. Louis, MO, USA) total end concentration 25 mM |
| WME + Dextran40 | WME + 10 μG/mL ciprofloxacin + 25 μG/mL Amphotericin B + 3.5% Dextran40 (Sigma-Aldrich, St. Louis, USA) |
| WME + Fungi (0.25) | WME + 10 μG/mL ciprofloxacin + 25 μG/mL Amphotericin B (Fungizone) (Merck) |
| WME + Fungi (1) | WME + 10 μG/mL ciprofloxacin + 100 μG/mL Amphotericin B |
| RPMI | Roswell Park Memorial Institute 1640 (RPMI) (Gibco) + 10 μG/mL ciprofloxacin + 11 mM D-(+)-glucose solution |
| Blood | Diluted blood NMP perfusate (See Appendix A) |

## 2.5. Evaluation of Viability

Viability of PCKS were evaluated by assessing the ATP content as described before by De Graaf et al. 2010 [27]. A bioluminescence kit was used (Roche Diagnostics, Basel,

Switzerland). Luminescence was measured using a luminometer (Packard LumiCount, Downers Grove, IL, USA). The obtained ATP values were normalized against total protein content using the Pierce$^{TM}$ BCA protein assay kit. The final ATP content was expressed as pmol ATP/µG protein.

Lactate dehydrogenase (LDH) and aspartate aminotransferase (ASAT) levels in the medium, as a marker for renal injury, were determined in a routine fashion at the lab of clinical chemistry (University Medical Center Groningen (UMCG)).

### 2.6. Histological Analysis

PCKS were fixed in 4% formalin, embedded in paraffin wax, and cut into sections of 4 µm. Sections were stained using a conventional HE staining to visualize morphological features. Sections were scanned with a C9600 NanoZoomer (Hamamatsu Photonics, Hamamatsu, Japan) to obtain high-resolution digital data. Semi-quantitative scores were assigned to HE-stained sections in a blinded manner by two individuals, marking glomerular dilatation and structure, tubular dilatation and acute tubular necrosis. Scores ranged from 0–2, with 0 representing no damage, 1 representing mild damage, and 2 representing severe damage.

### 2.7. Determination of Fungal Infection

The type of fungal infection was determined in a routine fashion at the lab of microbiology (UMCG).

### 2.8. Statistical Analysis

Data were visualized and analyzed using GraphPad Prism 8.0 (GraphPad Software, San Diego, CA, USA). Values are shown as means with appropriate standard error of the mean (SEM) and as individual values. A Kruskal–Wallis analysis of variance combined with a Dunn's multiple comparison test was performed to analyze statistical differences between experimental groups. The cut-off for statistical significance was set for $p < 0.05$.

## 3. Results

### 3.1. HMP-O$_2$ as Cold Preservation Results in Significant Higher PCKS Viability

We first analyzed which cold preservation technique resulted in the most viable slices after 48 h of incubation (Figure 2A). After cold preservation and the slicing procedure, no differences in ATP levels were observed. After 48 h of incubation in WME with glucose, PCKS showed significant higher ATP levels after oxygenated HMP-O$_2$ compared to the SCS preservation ($p = 0.0467$). The tubular dilatation trended higher at T0, with more tubular dilatation in the HMP-O$_2$ group (Figure 2B), but it was not significantly different. Acute tubular necrosis scores fluctuated over time but were similar in both groups after 48 h incubation (Figure 2C,D). As the ATP levels were significantly higher in the HMP-O$_2$ group, oxygenated HMP was chosen as the standard cold preservation method for the respective experiments.

### 3.2. Porcine PCKS Remain Viable up to 72 h

Next, we analyzed whether porcine PCKS can be incubated for up to 72 h to observe biological processes for a more extended period. After 3 h of oxygenated HMP with UW-MP, ATP levels significantly increased compared to after 30 min of WIT ($p = 0.0168$) (Figure 3A). After the slicing process (T0), ATP levels remained similar to before slicing, and ATP levels were significantly higher compared to after 30 min of WIT ($p = 0.0118$). Slices remain viable at up to 72 h of incubation with WME, as ATP levels were significantly higher after 24, 48, and 72 h ($p = 0.0100$, $p = 0.0374$, and $p = 0.0197$, respectively) of incubation with WME compared to T0. Tissue integrity by means of histological staining was studied next (Figure 3B). Tubular damage significantly increased over time, as shown by tubular dilatation and acute tubular necrosis (Figure 3C,D). Glomerular dilatation significantly

decreased after 24 h of incubation, and no further differences were seen between the times of incubation. (Figure 3E).

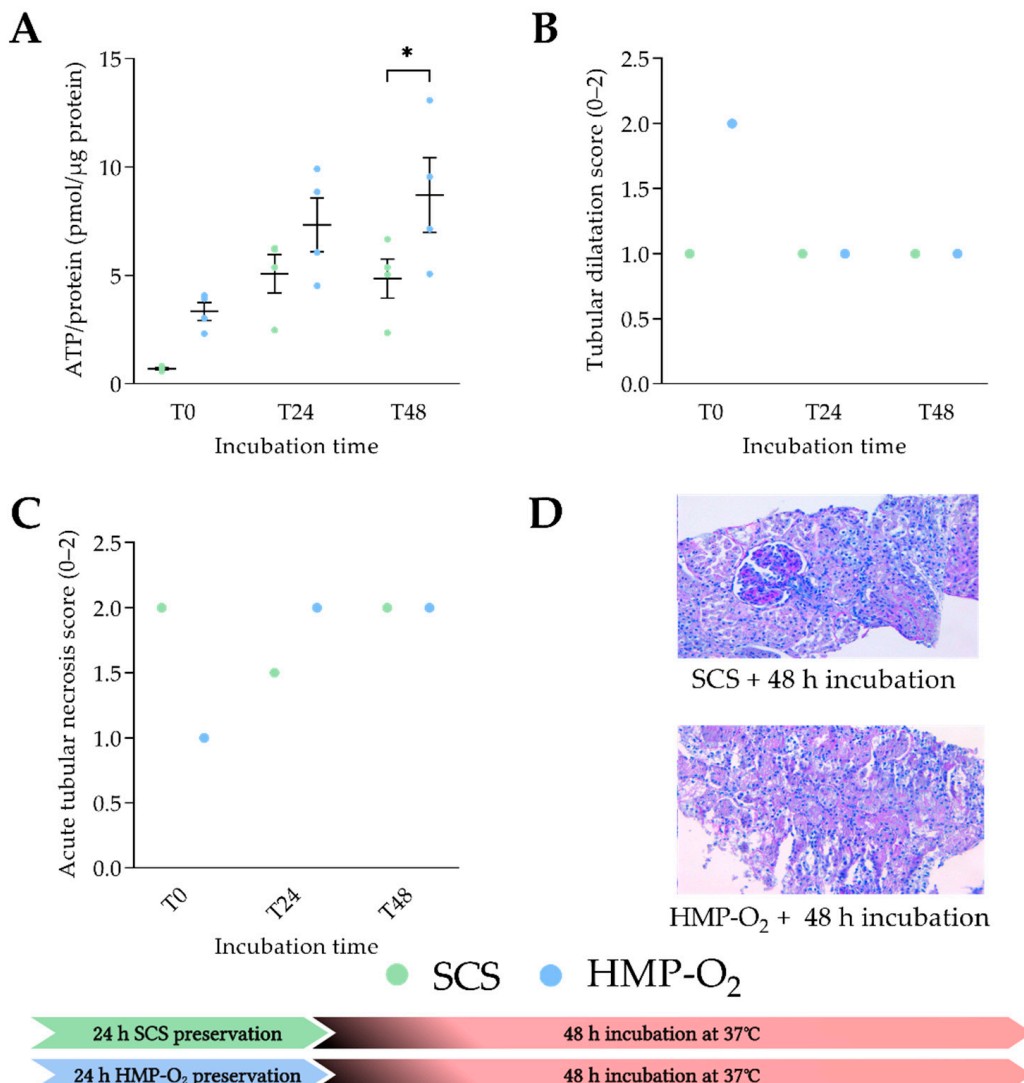

**Figure 2.** Tissue viability during PCKS incubation after cold preservation. (**A**) ATP levels of porcine kidneys that were preserved with oxygenated HMP (HMP-$O_2$) (●) ($n$ = 4) or static cold storage (SCS) (●) ($n$ = 4) for 24 h. (**B**) Histological tubular dilatation score of porcine kidneys. (**C**) Histological acute tubular necrosis score of porcine kidneys preserved with HMP-$O_2$ or SCS. (**D**) Histological images after 48 h of incubation of PCKS. T0 represents ATP/protein levels directly after slicing and T24 and T48 after incubation for 24 h and 48 h, respectively. * $p < 0.05$. SCS, static cold storage; HMP, hypothermic machine perfusion; ATP, adenosine triphosphate. The data are shown as mean $\pm$ SEM.

### 3.3. WME Provides the Most Beneficial Incubation Conditions

We then compared different medium compositions (Figure 4). After HMP-$O_2$ with UW-CS, ATP levels significantly increased compared to baseline when PCKS were incubated in WME, WME with added glucose, and RPMI ($p < 0.0001$, $p < 0.0001$, and $p = 0.0013$, respectively) (Figure 4A).

After HMP-$O_2$ with UW-MP, ATP levels increased significantly when PCKS were incubated in WME ($p = 0.0005$). When slices were incubated in a blood-based perfusate, ATP levels were significantly lower than in the WME group ($p = 0.0101$). The addition of Dextran40 resulted in lower ATP levels compared to WME alone; however, this was not a significant difference (Figure 4B).

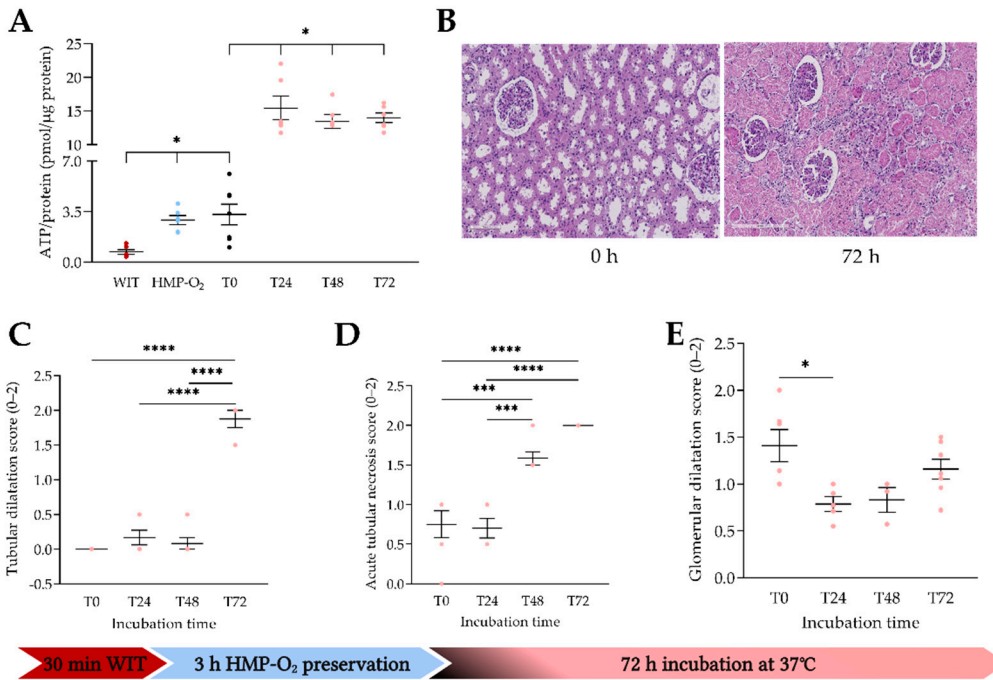

**Figure 3.** Tissue viability after slicing and up to 72 h of incubation. (**A**) ATP/protein levels after 30 min of WIT (●) (*n* = 6); porcine kidneys were preserved by means of oxygenated HMP (●) (*n* = 6) for 3 h, sliced (T0) (●) (*n* = 7), and incubated up to 72 h (●) (*n* = 6). (**B**) Histological images directly after slicing (T0) and after 72 h of incubation. (**C**) Histological tubular dilatation after incubation at different time points. (**D**) Histological acute tubular necrosis after incubation. (**E**) Histological glomerular dilatation. * $p < 0.05$, *** $p < 0.001$, **** $p < 0.0001$. WIT, warm ischemia time; HMP, hypothermic machine perfusion; ATP, adenosine triphosphate. The data are shown as mean ± SEM.

PCKS incubated in RPMI showed significantly more tissue injury in terms of LDH and ASAT production compared to WME (*p* = 0.0263 and *p* = 0.0088, respectively) (Figure 4C,E). No differences in injury markers were observed when glucose was added to the WME. Incubation of PCKS in a blood-based perfusate led to significant more LDH production compared to WME (*p* = 0.0219) (Figure 4D); however, no significant differences in ASAT levels were observed (Figure 4E). The addition of Dextran40 did not lead to more LDH or ASAT production compared to WME alone (Figure 4D,F).

### 3.4. PCKS Remain Viable under Both Hypothermic and Normothermic Conditions

To observe the effect of hypothermic incubation on viability, we incubated PCKS at 4 °C, 37 °C, and a combination of 4 °C and 37 °C (Figure 5). When slices were incubated at a normothermic temperature, ATP levels increased significantly compared to baseline (*p* = 0.0053). When PCKS were incubated under hypothermic conditions, or when slices were incubated at 4 °C for 24 h and then rewarmed to 37 °C for another 24 h, no significant increase in ATP levels were observed compared to baseline. When comparing 37 °C to 4 °C or 4 °C + 37 °C, no significant differences in ATP levels were observed (Figure 5).

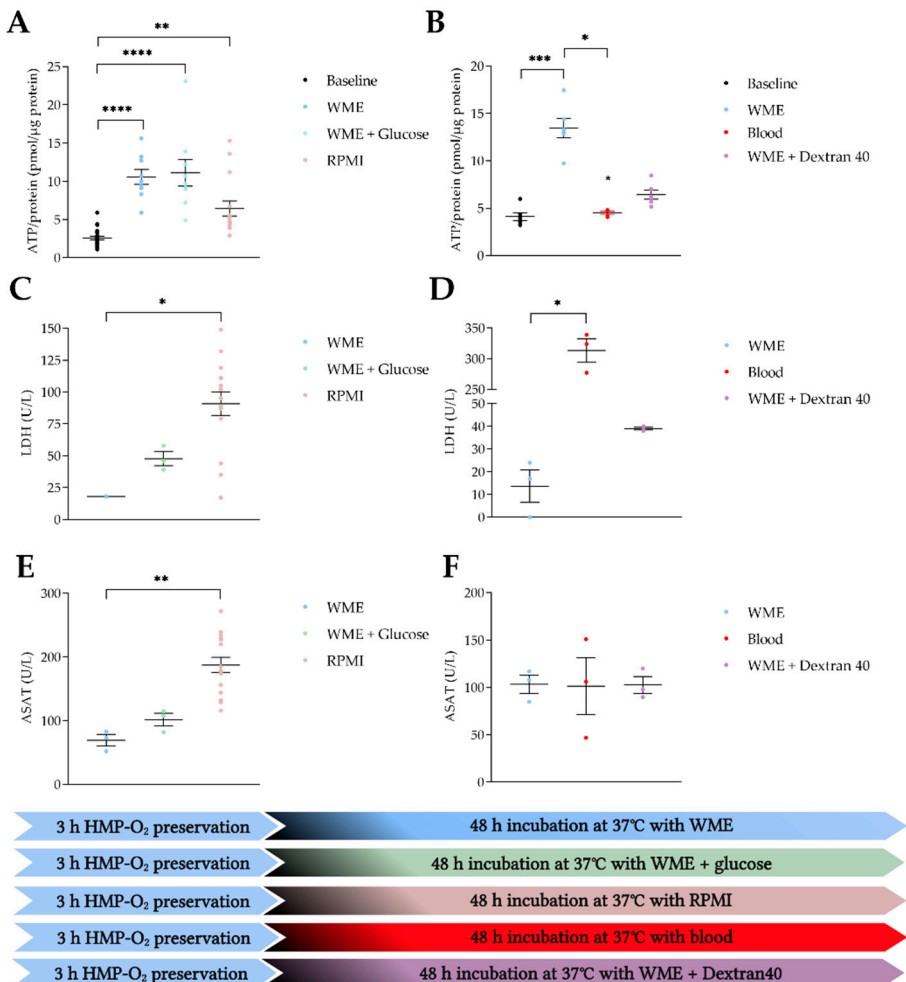

**Figure 4.** Viability and injury markers after 48 h of incubation using different medium compositions. (**A**,**C**,**E**) HMP-O$_2$ with UW-CS at baseline (●) (*n* = 24) and incubation with WME (●) (*n* = 9), WME with glucose (●) (*n* = 9), and RPMI (●) (*n* = 14). (**B**,**D**,**F**) HMP-O$_2$ with UW-MP at baseline (●) (*n* = 6) and incubation with WME (●) (*n* = 6), blood (●) (*n* = 5), and WME with Dextran40 (●) (*n* = 6). (**A**) ATP/protein levels after incubation with different media. (**B**) ATP/protein levels after incubation with different media. (**C**) Cumulative LDH levels after incubation with different media. (**D**) Cumulative LDH levels after incubation with different media. (**E**) Cumulative ASAT levels after incubation with different media. (**F**) Cumulative ASAT levels after incubation with different media. * $p < 0.05$, ** $p < 0.01$, *** $p < 0.001$, **** $p < 0.0001$. WME, William's medium E; RPMI, Roswell Park Memorial Institute; ATP, adenosine triphosphate; LDH, lactate dehydrogenase; ASAT, aspartate aminotransferase. The data are shown as mean ± SEM.

### 3.5. Amfotericine B Is Safe to Add to Prevent Fungal Infections

One of the obstacles during the incubation of the porcine PCKS was the occasional incidence of a fungal infection by *Candida guilliermondii* (Figure 6A). To prevent this, we analyzed whether Amphotericin B (fungizone) could be added to the medium without negatively affecting viability. WME and WME with added fungizone (25 μG/mL and 100 μG/mL) showed significant higher ATP levels compared to baseline (*p* = 0.0093, *p* < 0.0001 and *p* < 0.0001, respectively) (Figure 6B). Furthermore, no significant differences were observed after 48 h of incubation between the three experimental groups. Additionally, fungal infections in follow-up experiments were prevented.

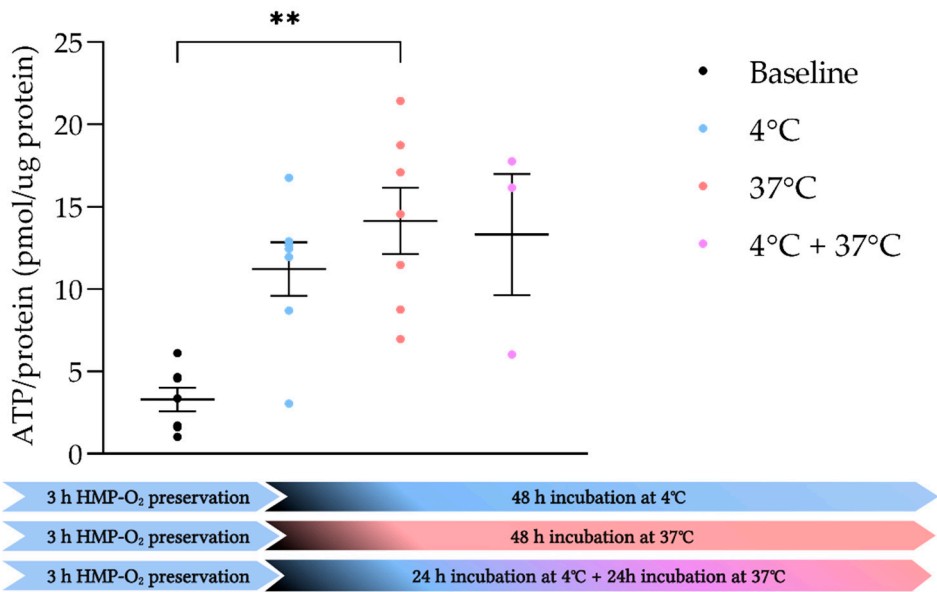

**Figure 5.** Incubation of slices at different temperatures. ATP/protein levels of porcine kidneys that were preserved with oxygenated HMP sliced at baseline (•) (*n* = 7) and incubated for 48 h in WME at 4 °C (•) (*n* = 7), 37 °C (•) (*n* = 7), or first 24 h at 4 °C and then rewarmed to 37 °C for 24 h (•) (*n* = 3). ** *p* < 0.01. ATP, adenosine triphosphate. The data are shown as mean ± SEM.

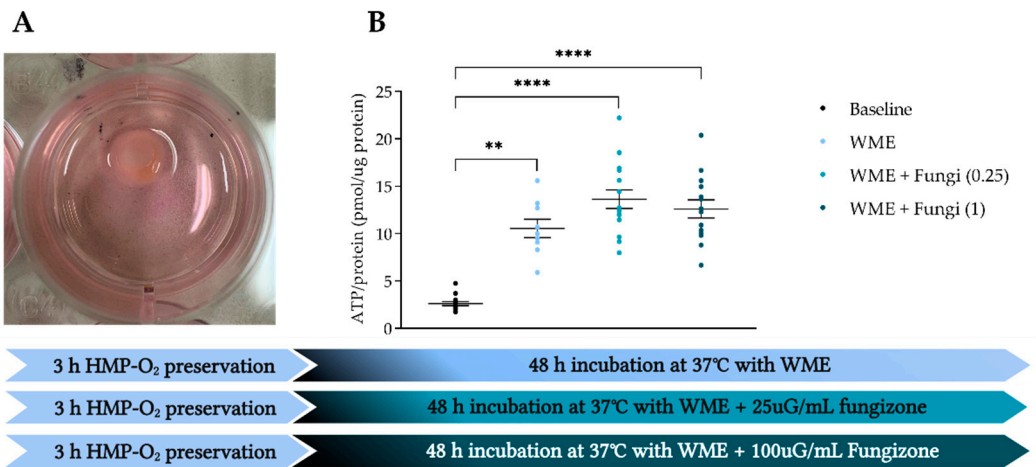

**Figure 6.** The addition of Amphotericin B (fungizone) to prevent fungal infections. (**A**) PCKS with a fungal infection by *Candida guilliermondii*. (**B**) Porcine kidneys were preserved with HMP, sliced at baseline (•) (*n* = 6), and incubated in WME (•) (*n* = 7) with 25 µg/mL fungizone (•) (*n* = 15) or 100 µg/mL fungizone (•) (*n* = 15) for 48 h. ** *p* < 0.01, **** *p* < 0.0001. WIT, warm ischemia time; ATP, adenosine triphosphate. The data are shown as mean ± SEM.

## 4. Discussion

These pilot studies aimed to investigate whether porcine PCKS in combination with cold preservation is suitable as an ischemia-reperfusion model for transplant-related research. The results reveal that oxygenated HMP leads to viable porcine PCKS for up to 72 h under normothermic and 48 h under hypothermic conditions. Furthermore, the results reveal that WME serves best to keep slices viable, and the addition of fungizone prevents the development of fungal infections without interfering with tissue viability.

### 4.1. Oxygenated HMP Provides an Excellent Foundation for Viable Porcine PCKS

When the kidney is transplanted after a period of cold ischemia, IRI may develop and could lead to DGF. HMP as a cold preservation technique is shown to reduce the risk of

developing DGF compared to SCS [10,11,28–30] and is, therefore, the standard care for all deceased donor kidneys in the Netherlands [31]. It is known that DGF kidney grafts fail to recover aerobic metabolism and therefore show ATP depletion after reperfusion [32]. We observed that PCKS were more viable in terms of ATP levels after preservation, and a similar degree of tubular damage was seen with oxygenated HMP compared to SCS (Figure 2). Additionally, we showed that, in terms of energy levels, HMP-$O_2$ results in viable PCKS for up to 72 h of incubation. However, after 72 h, the tubuli are damaged the most. (Figure 3). This is in line with studies that showed that HMP preservation improves the viability of renal tissue after normothermic reperfusion [13,33].

### 4.2. WME Best Supports the Renal Metabolism of Porcine PCKS

During NMP, a red-blood-cell-based perfusate is needed for optimal oxygen delivery to the tissue [34]. However, incubating slices in a diluted blood-based perfusate did not result in enough support for renal metabolism (Figure 4B). This could be due to the hemolysis, reflected by high LDH levels that were observed during incubation (Figure 4D).

As kidneys need colloid osmotic pressure for their filtration processes, we added Dextran40 to the incubation medium to test the effects on renal toxicity. Although these filtration processes are not active in PCKS, the adequate concentration could be translated to a perfusion model. The addition of a colloid during HMP increased renal metabolic activity during reperfusion [35]; however, we observed a decrease in PCKS viability with added Dextran40 (Figure 4B). This decrease could be explained by a decrease in oxygen and nutrients diffusion due to the higher viscosity of the incubation medium [35] or because the cells within the slices are more prone to stress caused by a higher colloid osmotic pressure, as they lack a glycocalyx [36].

Furthermore, WME provided enough nutrients for ATP production, and extra glucose did not increase viability, whereas incubation in RPMI resulted in lower ATP levels (Figure 4A). These results are in line with De Graaf et al. 2000 [37], who observed significantly higher cell proliferation in precision-cut liver slices incubated in WME compared to RPMI.

### 4.3. Porcine PCKS Can Be Incubated under Both Hypo- and Normothermic Conditions

As proposed before, HMP is a commonly used technique for preserving the kidney graft before transplantation. Therefore, the incubation of slices under both hypothermic and normothermic temperatures was investigated. With cold incubation of slices, prolonged hypothermic preservation can be studied efficiently. Although there is no active perfusion during PCKS incubation, the medium is oxygenated and agitated, resulting in oxygen and nutrient delivery to the renal cells, just like during machine perfusion. To our knowledge, hypothermic incubation and rewarming (4 °C and 37 °C) have not been studied before in PCKS but are promising methods to further explore in follow-up studies.

### 4.4. ATP as a Biomarker for Renal Tissue Viability

Kidneys are big consumers of oxygen, as renal cells require high energy levels due to their ATP-dependent functions such as reabsorption and secretion [38]. Therefore, the renal metabolic state is an excellent representation of kidney health [39]. To produce ATP and thus energy, the mitochondria need to work properly, and nutrients and oxygen are necessary to fuel these metabolic processes. It is also shown that renal ATP levels are inversely correlated to histological injury [40]. Therefore, ATP levels provide a great biomarker for kidney tissue viability [41].

### 4.5. Advantages and Disadvantages of Porcine PCKS

As already mentioned before, PCKS provide a platform to study organ-specific cellular mechanisms, as they maintain cellular heterogeneity and organ structure [23]. PCKS have been widely implemented for kidney research as a model to study the metabolism, toxicity, and efficacy of drugs [42–46]. The advantage is that multiple compounds or conditions using the same kidney can be studied, therefore keeping biological differences

minimal [47]. PCKS could provide a platform to investigate biopharmaceuticals that target IRI, inflammation, and fibrosis-related injury. These processes frequently lead to transplant-related complications, and no effective treatment is currently available.

As the kidney is not transplanted, it is challenging to mimic reperfusion using PCKS. During incubation, the tissue is rewarmed, and oxygen is reintroduced, presumably triggering cellular reperfusion injury. However, the strength of this model is to investigate ischemic damage caused by organ procurement and cold preservation.

Good kidney function is defined by good filtration and reabsorption [38]. Unfortunately, the glomerular filtration rate and sodium reabsorption cannot be measured, as PCKS do not produce urine. These parameters would have to be tested in follow-up studies using ex-vivo reperfusion studies. However, ex-vivo reperfusion for periods longer than 6 h remains a challenge.

Because of the size of the porcine kidney and the relative simplicity of the method, hundreds of slices can be produced and used to test different treatments under a variety of conditions. Slaughterhouse kidneys are commonly used to research renal transplant-related questions in NMP setups [48–54]. They provide an excellent alternative for laboratory animals and are more similar to human kidneys than any other species [55]. A limitation of the use of slaughterhouse kidneys is that these kidneys cannot be obtained in a (semi) sterile manner and are therefore more prone to infections. Our only observed infection during our experiments was a *Candida guilliermondii* infection, and adding Amphotericin B to the medium resolved this problem entirely, and no signs of nephrotoxicity were found.

### 4.6. Future Perspectives Porcine PCKS in Translation to NMP

Although NMP is a hot topic in the renal transplant field, it is clinically only scarcely implemented. One main obstacle of NMP remains the uncertainty of the needs of an isolated kidney during NMP [20,21]. PCKS could provide a platform to test different interventions such as electrolyte compositions, nutrients, and antioxidant supplementation and oxygen and carbogen concentrations simultaneously to better understand the underlying mechanisms of ischemia and how to repair this phenomenon, whereafter a translation to a perfusion study can be made. Furthermore, implementing human PCKS for transplant-related research would be of great interest to close the gap between interspecies differences.

### 4.7. Limitations

This pilot study consisted of small sample sizes, and the results are primarily based on viability analysis only. A larger sample size and more extensive analyses would be beneficial for follow-up studies. However, with these pilots experiments, we were able to illustrate the value of porcine PCKS as an ischemia-reperfusion model for transplant related research, which could be optimized in future studies.

**Author Contributions:** Conceptualization, L.A.v.F. and L.L.v.L.; methodology, L.A.v.F., H.G.D.L., I.A.M.d.G., P.O., L.S. and L.L.v.L.; data analysis, L.A.v.F. and L.L.v.L.; writing—original draft preparation, L.A.v.F. and L.L.v.L.; writing—review and editing, L.A.v.F., H.G.D.L., I.A.M.d.G., P.O. and L.L.v.L.; visualization, L.A.v.F. and L.L.v.L.; supervision, H.G.D.L. All authors have read and agreed to the published version of the manuscript.

**Funding:** This research received no external funding.

**Institutional Review Board Statement:** Not applicable.

**Informed Consent Statement:** Not applicable.

**Data Availability Statement:** All data available upon request.

**Acknowledgments:** We are grateful for butchery Kroon Vlees and thank Henk Luinge for providing kidneys for this research. We thank members of the Leuvenink, Olinga, and De Graaf group for their assistance with experiments and insightful discussions.

**Conflicts of Interest:** The authors declare no conflict of interest. The author Lorina Seras is an employee of MDPI, however she does not work for the journal *Transplantology* at the time of submission and publication.

## Appendix A

| Perfusate Composition: | |
| --- | --- |
| Heparinized and leukocyte-depleted autologous blood | |
| Red blood cells: | 360 mL |
| Plasma: | 475 mL |
| Amoxicillin/Clavulanic acid (1000 mg/200 mg) | 10 mG |
| 8.4% Sodium bicarbonate (B. Braun) | 15.25 mL |
| 5% Glucose (Baxter) | 12.50 mL |
| Dexamethasone (Centrafarm) | 8.3 mG |
| Mannitol (Baxter) | 10 mG |
| Creatinine (Merck) | 135 mG |
| Sodium nitroprusside (Merck) | 2.7 mG |
| Aminoplasmal (B. Braun) | 90 mL |
| Insulin (100 IU/mL) (Novorapid) | 0.186 mL |
| 250 µg/mL Amfotericine B (Fungizone) (Merck) | 1 mL |

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
