# Peer review of "Exploring Porcine Precision-Cut Kidney Slices as a Model for Transplant-Related Ischemia-Reperfusion Injury"

_2673-3943, doi:10.3390/transplantology3020015_

Round 1

Reviewer 1 Report

Please find the comments attached 

Author Response

See the attachement.

Reviewer 2 Report

This is an interesting approach to ischemia research in kidney transplantation, but it has moderate relevance to reperfusion injury. The methods proposed have its major use in studies of viability of tisue after ischemic insult, as well as testing of pharmacological treatment of the damaged tissue after CS vs HMP or NMP. The method have limited use to study outcome and functional aspects in the transplanted individual. as pointed out, the real problems arise when reperfusion starts, causing new damage to the organ that can not be studied easily using the proposed methods. I find the Dextran 40 and RBC parts not helpful. i do not see the clinical relevance. Detran 40 will stick to most surfaces and actually make oxygenation harder. Its extremely high colloid oncotic pressure will affect cells, probably even more in a slice setting. The major reason for using RBC in NMP, apart from oxygenation, is to use the buffering capacity. Since slices seem to do very well in culture medium, RBC will do very little. The authors need a more convincing physiological discussion to motivate the use. It has no relevance to a NMP setting in my opinion.

One of the strengths of the study, that the authors miss, is that the cells can survive for an extented time - much longer than in any whole organ setting. Despite the short warm ischemic setting, the resilience to ischemic insult seem to much better than what is normally considered.

I see the paper as nteresting, but it needs a rewrite with focus on the real use of the method, namely basic science of ischemia and possibly pharmacological manipulation in different settings.

Author Response

See the attachement.

Round 2

Reviewer 2 Report

I have no further comments.